# Learning Adaptive Tensorial Density Fields for *Clean* Cryo-ET Reconstruction

**Yuanhao Wang**    **Ramzi Idoughi**    **Wolfgang Heidrich**
King Abdullah University of Science and Technology (KAUST)
{yuanhao.wang,ramzi.idoughi,wolfgang.heidrich}@kaust.edu.sa

## Abstract

We present a novel learning-based framework for reconstructing 3D structures from tilt-series cryo-Electron Tomography (cryo-ET) data. Cryo-ET is a powerful imaging technique that can achieve near-atomic resolutions. Still, it suffers from challenges such as missing-wedge acquisition, large data size, and high noise levels. Our framework addresses these challenges by using an adaptive tensorial-based representation for the 3D density field of the scanned sample. First, we optimize a quadtree structure to partition the volume of interest. Then, we learn a vector-matrix factorization of the tensor representing the density field in each node. Moreover, we use a loss function that combines a differentiable tomographic formation model with three regularization terms: total variation, boundary consistency constraint, and an isotropic Fourier prior. Our framework allows us to query the density at any location using the learned representation and obtain a high-quality 3D tomogram. We demonstrate the superiority of our framework over existing methods using synthetic and real data. Thus, our framework boosts the quality of the reconstruction while reducing the computation time and the memory footprint. The code is available at https://github.com/yuanhaowang1213/adaptivetensordf.

## 1 Introduction

Tilt-series Cryo-ET is an important imaging tool used in a variety of fields, such as structural biology [5] and material science [48]. It consists of reconstructing a 3D tomogram from captured 2D projections of the scanned sample. In cryo-ET, due to hardware constraints, the scanning process is performed from a limited angular range, typically less than $120°$. The tomographic reconstruction with missing-wedges is known to be a challenging task. In addition, sample motion and deformation that occurs during the acquisition step, induce misalignment between the projections, which deteriorates the reconstruction quality. Also, cryo-ET requires a large amount of data, which makes the reconstruction a resource-intensive task, both in terms of computational time and memory. Last but not least, to prevent the sample from getting damaged, the scanning is performed with low-intensity electron beams. This results in high level noise in the captured projections.

In the literature, several approaches have been proposed to overcome some of these challenges. The misalignment and the motion can be corrected using marker tracking [28, 35, 49], or even by tracking specific features between the projections [10, 19]. On the other hand, the high level noise was initially reduced by applying image-based or volume-based denoising approaches on the projections or the tomogram respectively. Recently, Bepler et al. [7] proposed to reconstruct two copies of the tomogram using odd/even projections only. Then, using Noise2Noise [29] technique they learn a denoised version of the tomogram. Kniesel et al. [26] combines a noise model for Scanning Transmission Electron Microscopy (STEM) and an implicit 3D shape representation in a differentiable framework to denoise the reconstructed data.

37th Conference on Neural Information Processing Systems (NeurIPS 2023).

In this work, we propose a new framework for cryo-ET that jointly reconstructs and denoises the tomograms. This framework is based on a quadtree structure that we use to define an adaptive Tensorial Density Fields (**TensorDF**) representation. Each node of the quadtree is featured with a tensorial density fields to represent the tomogram's density at its associated region. The loss function that guides the learning process is composed of four terms: a differentiable tomographic image formation model, a total variation term that encourages smoothness in the reconstruction, a boundary consistency constraint that enforces agreement between the adjacent nodes, and an isotropic Fourier prior that penalizes directional artifacts and helps in denoising the reconstructed tomogram.

The evaluation of our approach on synthetic and real datasets shows considerable improvements comparing to state-of-the-art reconstruction methods, including some recent approaches based on neural representations. The main contributions of our work are summarized as follows:

- We propose an adaptive tensorial density fields representation, based on a quadtree structure, well-suited to large datasets. This allows us to achieve a breakthrough in 4K-resolution tomography reconstruction in less than a day.

- We propose a novel isotropic Fourier prior to remove the directional artifacts, and reduce the noise level.

## 2   Related Work

**Computed tomography (CT).**   Cryo-ET is one of the various types of tomographic inverse problem, which reconstructs a density volume of the scanned object from captured projections from different views. Analytical reconstruction algorithms, such as Filtered Back-Projection (FBP) [13] and Weighted Filtered Back-Projection (WFBP) [43] are commonly used to solve this problem. These approaches produce fast and accurate reconstructions but require a considerable number of projections that are uniformly sampled in the angular space. Algebraic Reconstruction Technique (ART) [18] and its variants, like the Simultaneous Algebraic Reconstruction Technique (SART) [2], solve the tomography problem in an iterative way. These methods are well-suited to challenging CT reconstructions (sparse view, missing wedges, noisy projections, etc.), since they can be combined with different regularizers like the total variation into optimization frameworks [21, 39, 23]. However, these approaches suffer from high computational requirements and fastidious hyper-parameter tuning. Learning-based approaches have been introduced in CT to improve reconstruction quality through pre-processing projections [3, 17, 52], post-processing the reconstructed tomogram [40, 32], or using neural networks to learn a differentiable reconstruction operator [1, 25, 20]. More recently, combining Deep Image Prior [4, 6] and implicit representations [53, 46, 45] with traditional reconstruction approaches has produced unprecedented reconstruction results. Nevertheless, these approaches are not well-designed to overcome the several challenges of cryo-ET, mentioned in Section 1, especially the high level noise.

**Cryo-ET data Denoising.**   Cryo-ET acquisition uses low-dose beams to avoid damaging the samples, but this results in capturing noisy projections [42]. In the literature, several models have been proposed for the cryo-ET noise, such as the additive white Gaussian noise [7, 56], or a Poisson-Gaussian noise [55]. Most existing denoising approaches are applied either before or after the reconstruction step [14], which is performed using classical algorithms like WFBP. The pre-reconstruction approaches aim to denoise the 2D projections using denoising algorithms such as the bilateral filter [24], the non-local means filetering [51], the wavelet shrinkage filter [22], and deep learning-based techniques [9]. On the other hand, post-reconstruction denoising is applied directly to the tomograms to save the linearity between captured projections and reconstructed volume. This approach is unlikely to introduce new artifacts or suppress existing features. Total variation [54], and BM4D [33] were the state-of-the-art in 3D denoising, for a decade. However, these approaches require considerable computational resources. In structural biology, subtomogram averaging [8] is a specific denoising technique for data containing several copies of the same molecule. Recently, learning-based methods gained great success in denoising tasks. Specifically, unsupervised methods like Noise2Noise [29], Noise2Void [27], present a great potential for tomograms denoising, given the absence of ground truth data in cryo-ET field. Thus, the Topaz algorithm [7] produces clean tomograms by leveraging the Noise2Noise concept to train on pairs of noisy tomograms reconstructed from odd/even projections. Kniesel et al. [26] proposed to jointly learn a model for 2D sensor noise and a 3D implicit representation of the scanned sample. However, this approach depends highly on

the noise level. In our approach, the denoising task is performed during the reconstruction process through the proposed loss function and the parametrization of our adaptive representation.

**3D Neural representation.** Neural representation has seen impressive and rapid development in the last few years [47]. The Neural Radiance Fields (NeRF) [36] use Multi-Layer Perceptron (MLP) networks to learn a mapping between spatial coordinates and physical scene properties (e.g., density field, color). However, NeRF-like approaches, also known as neural representations, suffer from long training times and slow rendering. Techniques such as octree structures [31, 15, 45, 50], multi-scale network architecture [34], network factorization [44], caching [16], multi-resolution hash encoding [37], and tensorial-based representation [11] have been applied to handle this issue. Some works adapted these neural representations to solve CT reconstruction with impressive results [46, 53, 45, 26]. In this work, we leverage a quadtree structure to build an adaptive tensorial-based representation. We also incorporate several regularizers in our loss function to handle the high level of noise in cryo-ET data. In addition, our method can deal with large datasets (4K resolution) in an unprecedented computational time.

## 3 Methodology

### 3.1 Overview

In this section, we introduce our approach, **Adaptive TensorDF**, which leverages a quadtree structure to create a multi-scale and effective tensor-based representation for reconstructing noisy tilt-series cryo-ET tomograms. Our framework represents the scanned sample's density field as a 3D continuous neural field. Since the cryo-ET tomograms have a low extension along the z-axis, we use an adaptive quadtree structure to partition the volume instead of an octree. Inside each node of the quadtree, the density field is represented using a tensorial representation. Our framework is composed of three steps: quadtree update, optimization of the tensorial representation, and 3D volume reconstruction. These steps are depicted in Figure 1. The quadtree update step involves optimizing the quadtree structure to achieve the best partition of the reconstructed volume. This is done by encouraging uniform nodes while limiting their total number. In the second step, we optimize a loss function to build a tensor-based representation of the density field inside each node. The proposed loss function incorporates a data-fidelity term and three regularizer terms, which we will detail in the following. The reconstruction step involves uniform sampling of the volume in the region of interest (ROI) and querying the density values at the sampled positions.

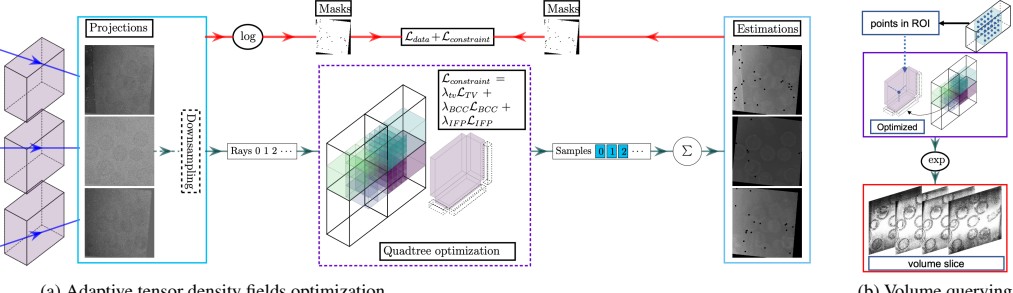

(a) Adaptive tensor density fields optimization.      (b) Volume querying.

Figure 1: Our framework is composed of two steps: During the training step (a) we first update the quadtree structure using downsampled projections. Then, we freeze that structure and update the tensor representation in each node using the original noisy projections. The second step is the volume querying (b), where we uniformly sample the ROI, and use the learned representation to estimate the densities at the selected positions.

### 3.2 Image formation model

Cryo-electron tomography is a classical tomographic reconstruction problem, where the projection image captured by the sensor corresponds to the integration of density along the rays between the source and the detectors in the log space. After discretization, the image formation model for a given ray, can be written as as follows:

$$\mathbf{b}_i = \mathbf{A}_i \mathbf{x} + \mathbf{n}_i \tag{1}$$

where $\mathbf{A}_i$ represents the Radon transform operation along the ray $i$. $\mathbf{b}_i$ and $\mathbf{n}_i$ correspond to the intensity measured by the detector $i$ and its associated noise, respectively. $\mathbf{x}$ is the 3D density vector that we would like to reconstruct. As discussed in Section 1, the noise in real Cryo-ET acquisitions is complex and difficult to model, especially after applying the different pre-processing steps such as the intensity correction and the motion correction. For the sake of simplicity, we assume in this work a Gaussian white noise in the pre-processed projections. By regrouping all the captured rays, we deduce the data-fidelity loss from the Equation 1:

$$\mathcal{L}_{data}(\mathbf{x}) = \frac{1}{2}\|\mathbf{K}\,(\mathbf{A}\mathbf{x} - \mathbf{b})\,\|_2^2 \qquad (2)$$

where $\mathbf{K}$ is a binary mask to disable rays intersecting with the fiducial markers. This mask is introduced to suppress the projection artifacts caused by those markers.

### 3.3 Coordinate-based representation (CBR)

In tomography applications, the coordinate-based networks have been proposed to map the 3D spatial coordinates inside the volume of interest to the density field. This mapping is given by:

$$f_\phi : \mathbf{p}_i \to \mathbf{x}_i \quad \text{with} \quad \mathbf{p}_i \in \mathbb{R}^3, \mathbf{x}_i \in \mathbb{R} \qquad (3)$$

where $\mathbf{p}_i$ and $\mathbf{x}_i$ are respectively the 3D coordinate in the volume to be reconstructed and the corresponding density. $f_\phi$ corresponds to the mapping function that should be optimized. With the CBR, the captured projections are estimated by sampling positions along rays, applying the mapping to each sample, and then summing their contribution using the Radon coefficients. In common implicit representation [36, 26], $f_\phi$ is chosen to be a fully connected MLP, as illustrated in Figure 2-(a). However, this representation is unsuitable for large volume sizes, as encountered in cryo-ET. To address this limitation, works like KiloNeRF [44], ACORN [34], and NeAT [45] suggested partitioning the volume of interest using uniform blocks or multi-scale octree-based structures, and assigning a smaller MLP or decoder to each partition (block or octree node) for a local density field representation. In our approach, shown in Figure 2-(b), we partition the ROI using a quadtree structure. Indeed, cryo-ET data have limited extension in the z-axis. Therefore, we base our representation on an adaptive quadtree structure of the x-y plan that we extend in the z-axis. For each quadtree node, we use a tensorial-based representation. Thus, for a given 3D point in the region of interest $\mathbf{p}_i$, we represent the density field using a sum of vector-matrix outer products, as follows:

$$f_\phi(\mathbf{p}_i) : \mathcal{D}\Big( \sum_{r=1}^{R} \mathcal{V}_r^X(\mathbf{p}_i)\mathcal{M}_r^{Y,Z}(\mathbf{p}_i) + \mathcal{V}_r^Y(\mathbf{p}_i)\mathcal{M}_r^{X,Z}(\mathbf{p}_i) + \mathcal{V}_r^Z(\mathbf{p}_i)\mathcal{M}_r^{X,Y}(\mathbf{p}_i) \Big) \to \mathbf{x}_i \qquad (4)$$

where $\mathcal{D}$ is a one-layer decoder network, that converts encoded features to the output density. $\mathcal{V}_r^X$, $\mathcal{V}_r^Y$, and $\mathcal{V}_r^Z$ correspond to factorized vectors of the three modes ($X$, $Y$, and $Z$) for the $r^{th}$ component of the tensor decomposition. Similarly, $\mathcal{M}_r^{Y,Z}$, $\mathcal{M}_r^{X,Z}$, and $\mathcal{M}_r^{X,Y}$ are matrix factors for the two denoted modes: $(Y, Z)$, $(X, Z)$, and $(X, Y)$, respectively. $R$ is the rank of the representation, which is a hyperparameter to be tuned according to the complexity of the reconstructed sample. Indeed, selecting a small rank can help denoise the tomogram, but it may lose some detailed features of the scanned sample.

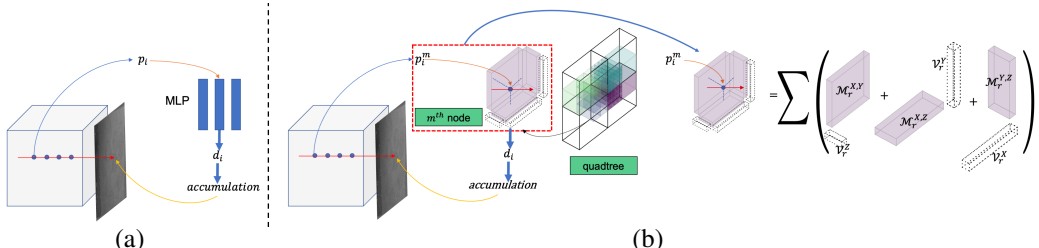

(a)  (b)

Figure 2: In (a), we show the application of neural radiance field to tomography reconstruction, while in (b), we illustrate the general idea of our approach. Where we replaced the MLP by an adaptive differentiable tensor fields for faster optimization and better feature recovery.

The main advantage of the tensor representation is the huge reduction of the number of parameters needed to represent the volume. Consequently, this representation is well-suited to reconstruct the large-size volume that we commonly encounter in cryo-ET datasets.

Furthermore, it is important to take into account the contribution of the regions outside the ROI in the projection process, to ensure a more accurate output. However, there is no need to have a high-quality representation in such regions. We propose to use the same representation detailed in Eq. (4), but with smaller size tensors.

## 3.4 Regularizations

**Total Variation.** The Total Variation (TV) loss is commonly used in traditional tomographic reconstruction as a spatial regularizer to improve the reconstruction quality. The use of TV in neural fields representation approaches is not always straightforward, and could increase the computational complexity of the reconstruction. [45] proposed to compute the TV loss in the features space, before applying the decoder to get the density. [11] used this loss on the vector and matrix factors, to handle noise and outliers issues in regions with fewer observations. In our implementation, we also apply the TV prior on the vector and matrix factors of enabled quadtree node in the ROI. Then we average all the contributions of those nodes to compute the TV loss, which could be expressed as follows:

$$\mathcal{L}_{TV} = \frac{1}{N_{EN}} \sum \alpha \cdot \text{mean}\left(\|\nabla \mathcal{V}(\mathbf{p})\|\right) + \text{mean}\left(\|\nabla \mathcal{M}(\mathbf{p})\|\right) \tag{5}$$

where $\alpha$ is a scaling factor. We find empirically that $\alpha = 0.1$ produced the best results. $\nabla \mathcal{V}(\mathbf{p})$ and $\nabla \mathcal{M}(\mathbf{p})$ represent the gradient of the vector and matrix factors respectively. $N_{EN}$ is the number of enabled nodes of the quadtree.

**Boundary Consistency Constraint.** In our proposed framework, each node of the quadtree has its own feature representation and is optimized separately. This will inevitably introduce discontinuity artifacts in the final reconstruction. To address this issue, we propose a Boundary Consistency Constraint (BCC) that penalizes the discrepancy between the features obtained from the tensor representation of neighboring nodes for sampling points on their shared edge. Our proposed BCC loss is given by:

$$\mathcal{L}_{BCC} = \sum_{(n,m) \in O_b} \text{mean}\left(\sum_{\mathbf{p} \in \cap_{n,m}} \|f_\phi(\mathbf{p})_m - f_\phi(\mathbf{p})_n\|\right) \tag{6}$$

where $O_b$ refers to all pairs of neighboring quadtree nodes, $\cap_{n,m}$ is the set of sampling points on the boundary surface between nodes $n$ and $m$, $f_\phi(\mathbf{p})_m$ and $f_\phi(\mathbf{p})_n$ are the densities evaluated at the point $\mathbf{p}$ using the tensor representation of the nodes $m$ and $n$, respectively.

**Isotropic Fourier Prior.** By combining the quadtree structure with the tensor density fields representation, we reduced the number of parameters needed to represent the 3D volume, as well as the reconstruction time. However, a downside of the low rank tensor representation is that it favors reconstructions with structured artifacts in the form of axis aligned streaks.

In the Fourier space, these artifacts manifest themselves as high peaks along the vertical and horizontal directions. To solve this issue, we introduced in our loss an Isotropic Fourier Prior (IFP), to penalize such peaks in Fourier space. This constraint aims to limit the difference between the horizontal and vertical frequencies and the other frequencies in the Fourier domain. To do this, we first calculate the mean amplitude for each ring of the Fourier domain, which represents a given spatial frequency for all possible directions. Then, we apply a penalty to the horizontal and vertical frequencies that are much higher than this mean amplitude. Furthermore, we apply a weighting coefficient to penalize more the high frequencies that correspond mainly to the noise. In our implementation, we sample the ROI coarsely, and we query the volume at those samples. Then, we compute the Fourier transform slice by slice. Our proposed IFP loss could be expressed as follows:

$$\mathcal{L}_{IFP} = \sum_{s,\zeta} \text{w}(\zeta)\Big(\big|F_\phi(0,\zeta) - \text{mean}\left(F_\phi(u,v)_{|(u^2+v^2=\zeta^2)}\right)\big| +$$
$$\big|F_\phi(\zeta,0) - \text{mean}\left(F_\phi(u,v)_{|(u^2+v^2=\zeta^2)}\right)\big|\Big) \tag{7}$$

where $F_\phi$ is the Fourier transform of $f_\phi$ computed on the slice $s$. $u$, $v$, and $\zeta$ are spatial frequencies. $\text{w}(\zeta)$ is the weighting coefficient given by:

$$\text{w}(\zeta) = \frac{1}{\exp\left(\zeta_0 - \zeta\right) + 1} \tag{8}$$

$\zeta_0$ is a frequency parameter, to control the weight of the loss according to the frequency. Empirically, we found that $\zeta_0 = 0.2$ provides an optimal weighting for the isotropic Fourier prior.

### 3.5 Adaptive tensor density field optimization

We have defined our tensor density representation and the different losses used for our optimization. By combining all those terms, our loss is given by:

$$\mathcal{L} = \mathcal{L}_{data} + \lambda_{TV} \cdot \mathcal{L}_{TV} + \lambda_{BCC} \cdot \mathcal{L}_{BCC} + \lambda_{IFP} \cdot \mathcal{L}_{IFP} \tag{9}$$

where $\lambda_{TV}$, $\lambda_{BCC}$, and $\lambda_{IFP}$ are the weights of the three regularizers defined previously. In the following, we describe the key elements of our framework's training step: updating the quadtree and sampling along the rays.

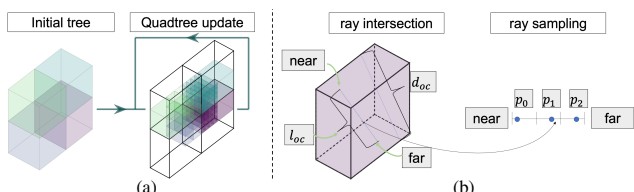

Figure 3: Illustration of (a) the quadtree updating process, and (b) the ray sampling strategy.

**Quadtree update.** As mentioned previously, we build our tensor representation using a quadtree structure, illustrated in Figure 3. First, we initialize our quadtree structure from the ROI, and we disable the nodes outside this region. Then, we sample uniformly the volume of each node, and compute the standard deviation (STD) of the density values of the sampled positions, to define the quadtree update loss. A high STD in a given node indicates that it contains fine details. Therefore, this node is more likely to be split into four child nodes. On the other hand, a node with low STD probably contains less features and represents uniform region of the scanned sample. Such node has higher chance to be merged or kept the same. We use a mixed-integer program (MIP) [41] to optimize this process, where, in each iteration, the nodes are either merged, split, or kept the same according to the update loss. During this optimization, the total number of nodes should stay lower than a fixed limit.

Furthermore, we use a coarse-to-fine strategy to speed up the quadtree update and deal with the high noise level. We first update the quadtree using down-sampled projections. During this step, we also update the tensor representation inside each node. This strategy reduces the impact of noise on the quadtree update. After some epochs, we keep the quadtree structure fixed and continue optimizing the tensor representation of each node using the original projections.

**Ray sampling.** During the optimization, we sample each ray to get a list of 3D positions to be used for the density integration and the loss evaluation. The applied sampling is not uniform, but follows a stratified random sampling strategy that takes into account on the quadtree structure. For each node that the ray goes through, we pick $N_q$ 3D positions on the ray, randomly sampled from $N_q$ uniform segments of the ray. We define $N_q$ as follows:

$$N_q = \lceil N_{\max} \frac{l_q}{d_q} \rceil \tag{10}$$

where $N_{\max}$ is a hyperparameter corresponding to the maximum number of samples per node, $l_q$ is the length of the ray inside the quadtree node $q$, and $d_q$ is the length of the diagonal of $q$.

## 4 Experiments

We fully implemented our framework in C++, which helps in speeding-up the reconstruction process. Implementation details could be found in the Supplement.

We designed a series of experiments to showcase the effectiveness of our framework on both synthetic and real captured datasets. We compare the performances of our approach to different baseline methods: (1) **SART+TV**, a well established iterative reconstruction technique **SART** combined with a total variation prior. (2) **Kniesel et al.**, an implicit neural representation for cryo-ET proposed by [26]. (3) **I-NGP**, a reimplementation of Instant-NGP [37] for cryo-ET reconstruction. (4) **TensoRF**, a reimplementation of Tensor Radiance Fields [11] for cryo-ET reconstruction. For a fair comparison, all the output densities are normalized into $[0, 1]$.

### 4.1 Experiments on synthetic dataset

In our experiments, we used the synthetic dataset introduced in [26]. This dataset consists of randomly distributed ellipsoidal shells with random densities that mimic the density model of the ZIKV (i.e., Zika) virion at 15Å. The simulated tomogram is projected in the angular range of $[-70°, 69.5°]$, with an angular step of $1.5°$. Gaussian noise is then added to the projections and several noise levels have been tested. When the standard deviation of the noise exceeds 0.05, the simulated projections look visually close to the captured ones, in terms of noise. This synthetic dataset is used for parameter tuning and robustness evaluation of our approach to the noise level.

**Parameter tuning**    After several experiments on all the hyperparameters of our approach, it appears that the most important parameters are the tensor dimensions (the dimension of the matrix-vector factors) and the feature size. The choice of tensor dimensions affects the speed and quality of the training and reconstruction. Smaller dimensions lead to faster training and smoother reconstruction, but they may miss some fine details of the sample. Larger dimensions capture more details, but they may also introduce overfitting or noise, as the network tends to learn the noise after learning the structures (See the Supplement for visual comparison). The feature size also influences the training speed and the reconstruction quality. Smaller features can speed up the training, but they may lose some details. Larger features can preserve more details, but they may increase the computation cost. A balance between these trade-offs should be sought for a better recovery. To study the impact of these two parameters, we measured the 3D Peak-Signal-to-Noise Ratio (PSNR) and 3D Structural Similarity Index Measure (SSIM) of the reconstructed volume using different values of these parameters. The results are shown in Table 1 and Table 2.

<table>
<tr><td colspan="7" align="center">Table 1: Impact of the tensor dimensions.</td></tr>
<tr><td></td><td>10</td><td>14</td><td>18</td><td>22</td><td>26</td><td>30</td></tr>
<tr><td>PSNR</td><td>35.86</td><td>36.82</td><td>37.15</td><td>37.45</td><td>**37.50**</td><td>37.12</td></tr>
<tr><td>SSIM</td><td>0.769</td><td>0.797</td><td>0.803</td><td>**0.820**</td><td>0.747</td><td>0.729</td></tr>
</table>

<table>
<tr><td colspan="7" align="center">Table 2: Impact of the feature size.</td></tr>
<tr><td></td><td>4</td><td>8</td><td>12</td><td>16</td><td>20</td><td>24</td></tr>
<tr><td>PSNR</td><td>37.21</td><td>37.45</td><td>37.47</td><td>**37.86**</td><td>37.42</td><td>37.68</td></tr>
<tr><td>SSIM</td><td>0.788</td><td>0.820</td><td>0.819</td><td>**0.837**</td><td>0.822</td><td>0.827</td></tr>
</table>

From the analysis of Table 1 and Table 2, the best performances are obtained with a tensor dimensions and feature size equals to 22 and 16, respectively.

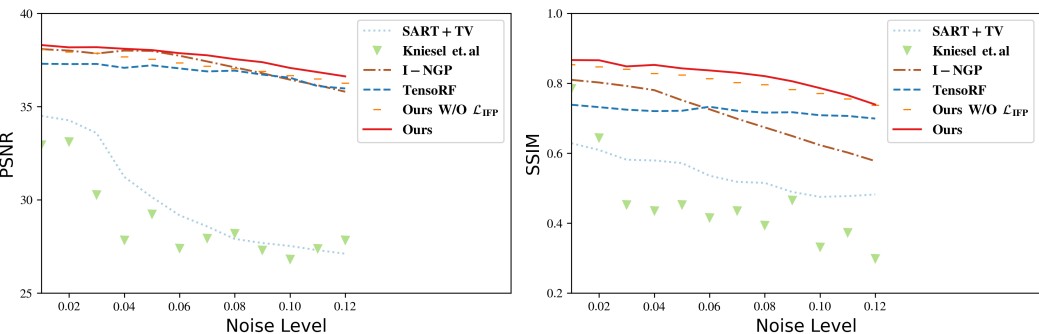

Figure 4: PSNR and SSIM comparison for different noise levels.

**Robustness to the noise level**    In this experiment, we simulate different noise levels, and compare our results with baseline methods. Figure 4 illustrates the PSNR and SSIM computed on the results of each methods for different level of noise in the range $[0.01, 0.12]$ for the standard deviation. Our approach consistently outperforms the other methods in terms of both PSNR and SSIM, regardless of the noise level. The neural representation based approaches are quite robust to the noise, since the performance do not drop consequently for larger level of noise. However, **Kniesel et al.** is not adapted to high noise situations, because it relies on the noise levels that it was learned from. In Figure 5, we show a slice visualization representing the reconstruction results using the different methods, from simulated noisy projections. The noise level have been selected to be 0.08, to illustrate the robustness to noise level of each approach. Two regions in the dataset are zoomed in to illustrate the detail recovery (red frame) and the denoising effect in uniform regions (blue frame). **SART+TV** shows a poor detail recovery and maintain a high noise level. **Kniesel et al.** approach is relatively

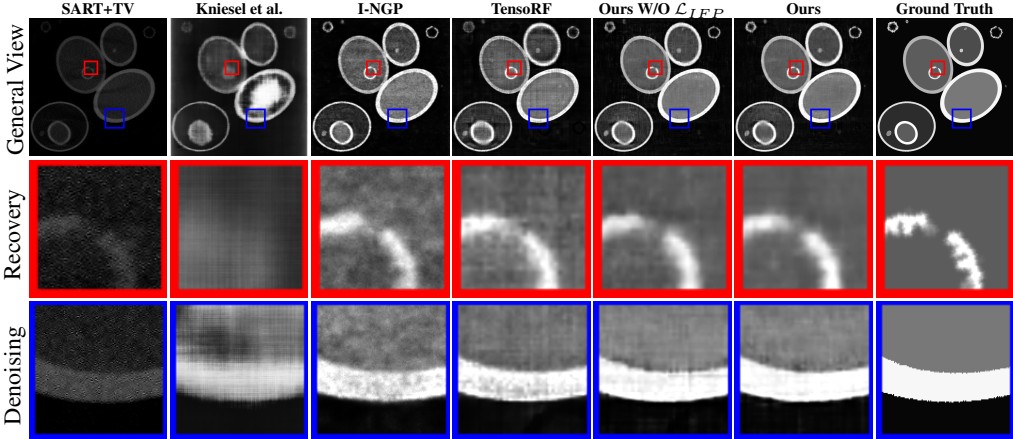

Figure 5: Reconstruction results with different methods from simulated noisy projections (Noise level=0.08). Zoomed regions show details recovery (red box) and denoising effect (blue box).

good in denoising uniform regions, but the details are not recover correctly. **I-NGP** and **TensorRF** approaches perform relatively good in both preserving details and reducing the noise. **TensorRF** introduces, however, some artifacts that we can notice in some ellipses in the *General View*. **Ours** visually has the best performances in both tasks. By comparing **Ours** and **Ours W/O** $\mathcal{L}_{IFP}$, we can see the impact of the Isotropic Fourier Prior in reducing artifacts and improving the denoising effect (More comparisons can be find in the Supplement).

## 4.2 Experiments on real dataset

We evaluated our method on tilt-series datasets from EMPIAR (Electron Microscopy Pilot Image Archive): EMPIAR 10643 [38], and 10761 [12]. The EMPIAR 10643 dataset is a cryo-ET acquisition of the HIV-1 GagdeltaMASP1T8I assemblies with an angular range of $[-60°, 60°]$ and an increment of $3°$. From this dataset, we reconstructed two different series (40 and 51) independently. The EMPIAR 10751 dataset corresponds to the cryo-ET acquisition of a HEK cell, with an angular range of $[-60°, 60°]$. The three used datasets are pre-processed using IMOD [35] for the projection alignment and contrast transfer function (CTF) correction. For a fair comparison, we normalize all the output densities to the range $[0, 1]$. In Figure 6, we show a comparison of the reconstruction results obtained with our approach and the four baselines methods, on EMPIAR 10643-40 and EMPIAR 10751 datasets. More results could be found in the Supplement.

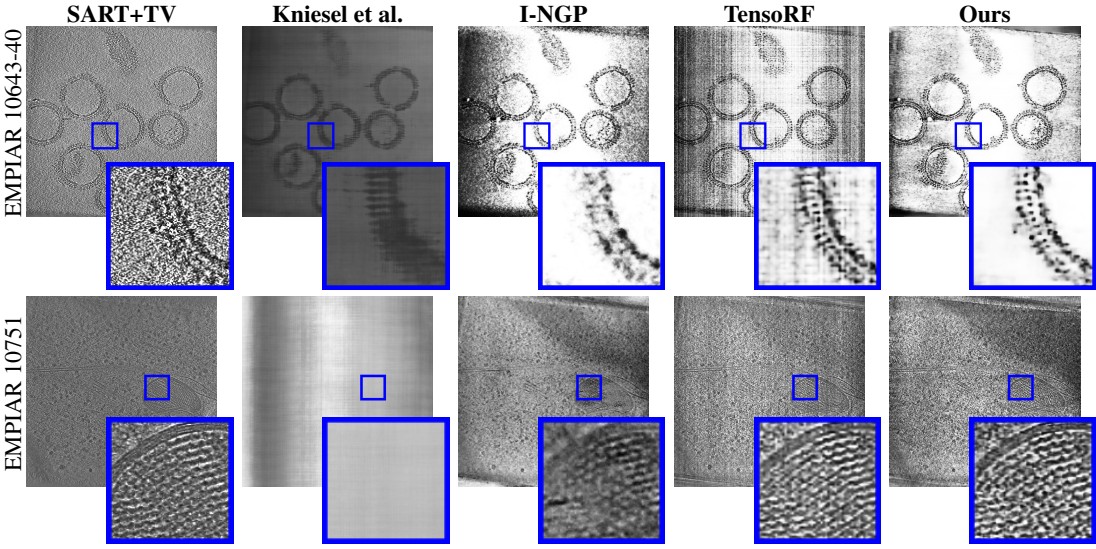

Figure 6: Reconstruction results of the HIV-1 (EMPIAR 10643) and a HEK cell (EMPIAR 10751).

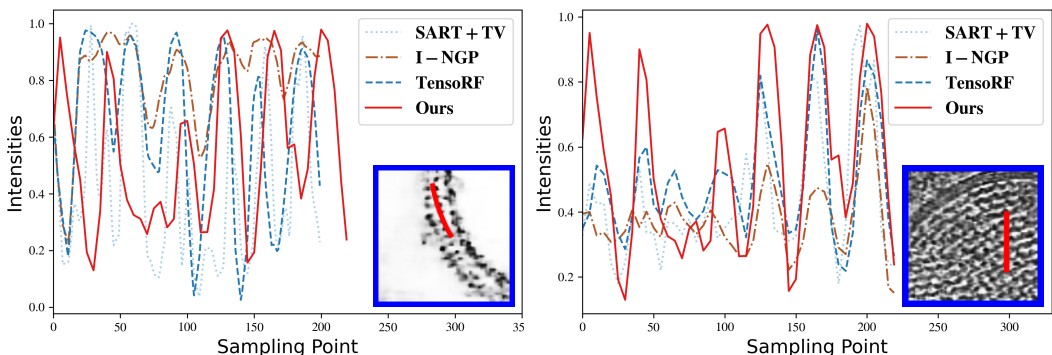

Figure 7: Intensity profile along the red line in the zoomed regions of the reconstruction of EMPIAR 10643-40 (Left), and EMPIAR 10761 (right) datasets.

**Evaluation of the denoising power**   In Figure 6, the first dataset shows our approach has the best denoising in uniform regions. Despite using the TV regularization, **SART+TV** still exhibit noise within uniform regions. Increasing the weight of the TV constraint to enhance the **SART+TV**'s denoising capabilities would result in the loss of main features. **Kniesel et al.** yields to an oversmoothed reconstruction, it could be caused by the difference between the datasets used for their learning of the noise statistic and our used dataset. **I-NGP** and **TensoRF** have a relatively good denoising. Some directional artifacts degrade the quality of **TensoRF**. They are due to the impact of dummy regions (outside the ROI), which are not well modeled with this approach. Besides its best denoising power, our approach is also the most effective in improving the contrast between the features and the background.

Table 3: Evaluation of the contrast enhancement (CNR) and the smoothing effect (ENL), (higher is better).

| Metric | Dataset | SART+TV | I-NGP | TensoRF | Ours |
|---|---|---|---|---|---|
| CNR ↑ | 10643-40 | 0.0618 | 0.5455 | 0.4123 | **0.6808** |
| | 10643-51 | 0.0578 | 0.4981 | 0.4801 | **0.5367** |
| ENL↑ | 10643-40 | 8.143 | 52.231 | 48.293 | **114.400** |
| | 10643-51 | 9.033 | 24.467 | 24.419 | **66.015** |

To assess quantitatively the denoising effectiveness of each approach, we performed a comparison using two statistical metrics: the Contrast-to-Noise Ratio (CNR), and the Equivalent Number of Look (ENL) [30]. The CNR metric measures the contrast between the region of interest and the uniform background. While, the ENL metric evaluates the smoothness in the uniform areas. We did not include the **Kniesel et al.** apprach in this comparison, since it does not perform well in the reconstruction of the real captured data (See Figure 6). The results shown in Table 3 confirm our qualitative observations by showing higher CNR and ENL values. Our approach enhances the contrast between feature regions and the background while producing smoother uniform regions. The table also indicates that the SART-TV result is the noisiest.

**Quadtree regularization**   The quadtree structure has more benefits than just speeding up the computation. It allows us to have more local matrix representations in the XZ and YZ planes, compared to TensoRF which can only have global planes for the whole scene. This can reduce the noise and misrepresentation in the reconstruction at a sacrifice of only a 50 percent reduction in parameter size. We compare our approach to: **TensoRF W/O** $\mathcal{L}_{TV}$, **TensoRF**, and **TensoRF W** $\mathcal{L}_{IFP}$ in Table 4 and Figure 8. Our method has better detail preservation and denoising. Our method achieves a significant improvement in quality.

Table 4: Comparison of the different method in terms of the contrast enhancement (CNR) and the smoothing effect (ENL).

| Metric | Dataset | TensoRF W/O $\mathcal{L}_{TV}$ | TensoRF | TensoRF W $\mathcal{L}_{IFP}$ | Ours |
|---|---|---|---|---|---|
| CNR ↑ | EMPIAR 10643-40 | 0.2884 | 0.4123 | 0.4470 | **0.6808** |
| | EMPIAR 10643-51 | 0.4530 | 0.4801 | 0.5218 | **0.5367** |
| ENL↑ | EMPIAR 10643-40 | 54.287 | 48.293 | 69.483 | **114.400** |
| | EMPIAR 10643-51 | 21.120 | 24.419 | 25.256 | **66.015** |

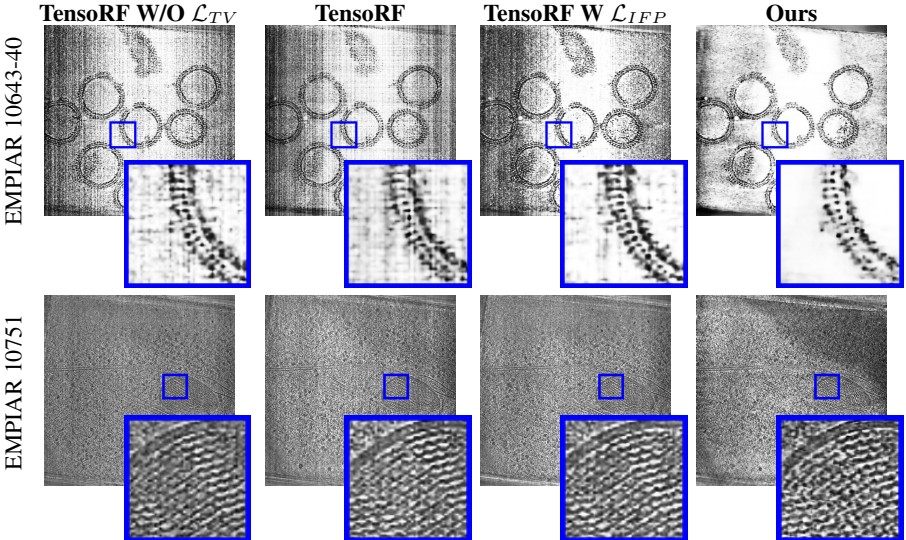

Figure 8: Reconstruction results of the HIV-1 (EMPIAR 10643) and a HEK cell (EMPIAR 10751).

**Detailed feature analysis** Figure 6 shows that our approach has the best feature recovery. Thus, for the EMPIAR 10643-40 dataset, our reconstruction allows a better resolving of the spike proteins. In addition, our reconstruction of the EMPIAR 10751 results in less blurry and more contrasted structures of the HEK cell. These observations are confirmed by the intensity profile in Figure 7, where our approach yields a more regular profile with an important difference between peaks (background regions) and valleys (features). On the other hand, all other methods show more intermediate peaks and valleys due to the residual noise in their reconstructions.

## 5    Conclusion and future work

In this paper, we introduced **Adaptive TensorDF** as a novel technique for fast and high-quality Cyro-ET reconstruction and denoising. Our technique leverages a quadtree structure to represent the density field using a vector-matrix factorized tensors representation. We optimize the quadtree structure and the tensor representation in a two-stage process, using the down-sampled and then the original projections. We also combine three priors with the tomographic formation model into the loss function: a total variation term, a boundary consistency constraint, and an isotropic Fourier prior.

Extensive experiments demonstrate that our technique outperforms the existing methods in terms of reconstruction quality and speed. It also involves a reasonable number of parameters (three times less than **TensoRF**). Besides, **Adaptive TensorDF** is scalable and efficient, and can handle 4k resolution projections, as shown in the Supplement, which is significant for high resolution cyro-ET dataset.

A possible direction for future work is to develop a joint alignment and reconstruction approach that can account for the misalignment between the projections and the volumes. This could improve the accuracy and robustness of our framework.

**Acknowledgement** The authors would like to thank the NeurIPS'23 reviewers for their constructive suggestions. This work was supported by King Abdullah University of Science and Technology as part of the VCC Competitive Funding as well as the CRG program.

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
