# Learning Adaptive Tensorial Density Fields for *Clean* Cryo-ET Reconstruction — Supplementary —

**Yuanhao Wang**     **Ramzi Idoughi**     **Wolfgang Heidrich**

King Abdullah University of Science and Technology (KAUST)

{yuanhao.wang,ramzi.idoughi,wolfgang.heidrich}@kaust.edu.sa

## A  Implementation details

We fully implemented our framework in C++. We ran the validation experiments of our framework on two workstations with the same CPU and memory characteristics: Intel(R) Xeon(R) Gold 6242 and 512 GB. One workstation is equipped with 4 Nvidia RTX 8000s 48G gpus and the other has 4 Nvidia 48G A6000 gpus. Both of these workstations were running on Ubuntu 18.04 LTS. We used only one GPU to run our experiments.

We used the root mean squared propagation optimizer (RMSProp), with a learning rate of $0.01$, and a decoder learning rate of $5 \cdot 10^{-3}$.

We defined one pass through all the available projections data as one epoch, and ran 30 epochs to ensure the optimization convergence. We followed the strategy described in Section 3.5 and optimized the quadtree and the tensor-based density fields for 15 epochs, then we fixed the structure and optimized only the density fields for another 15 epochs. During the first step we used downsampled projections (by a factor of $4$).

We set the maximum number of quadtree nodes to 70 in real datasets. The tensor dimension size depends mainly on the volume size to be reconstructed, and the scene complexity. Here, we set the quadtree size for the real scene as $64$, and the depth direction of the dimension size is calculated according to the ratio of the depth to the width or the height, as:

$$n_Z = n_X \cdot \frac{l_Z}{l_X} \tag{1}$$

where: $n_X$ and $n_Z$ are the tesnor dimensions in the X and Z-axis respectively. $l_X$ and $l_Z$ are the reconstructed volume size in the X and Z-axis respectively. The feature size was set to $16$ to get a better reconstruction as evaluated in Section 4.

After the parameter tuning search, we found $\lambda_{TV} \in \left[10^{-5}, 10^{-4}\right]$, $\lambda_{BCC} \in \left[1 \cdot 10^{-5}, 5 \cdot 10^{-5}\right]$, and $\lambda_{IFP} \in [0.01, 0.05]$ worked best.

## B  Additional experiments using synthetic datasets

The tensor dimensions and the feature size play an important role for the reconstruction quality as discussed in the main paper. In the Figure I, we illustrate how the tensor dimensions impact the reconstruction quality of the synthetic data. And in the Figure II, we show the impact of the feature size.

37th Conference on Neural Information Processing Systems (NeurIPS 2023).

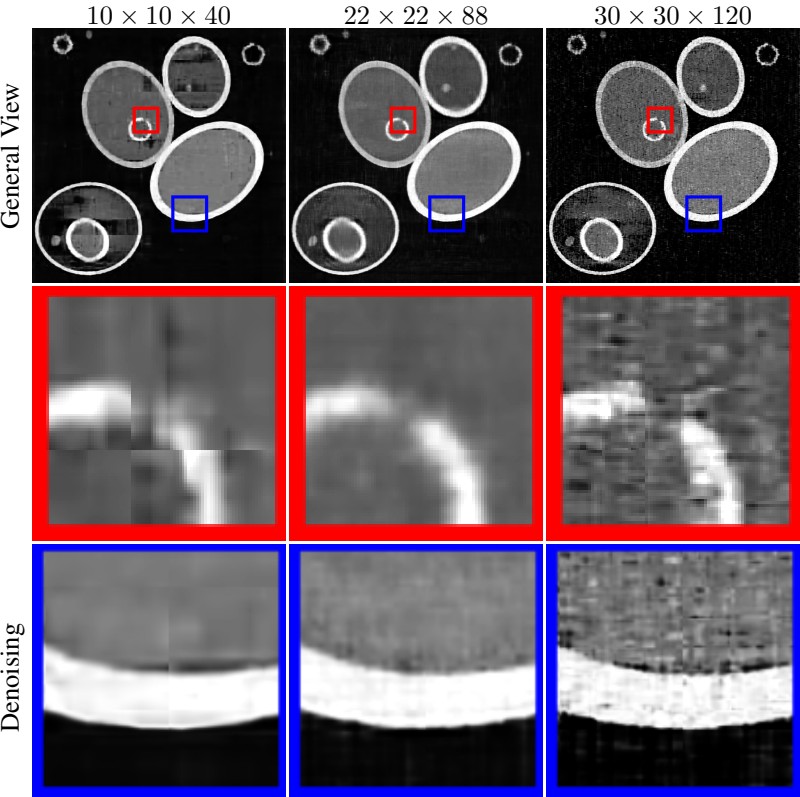

Figure I: Illustration of the impact of the tensor dimensions on the reconstruction quality.

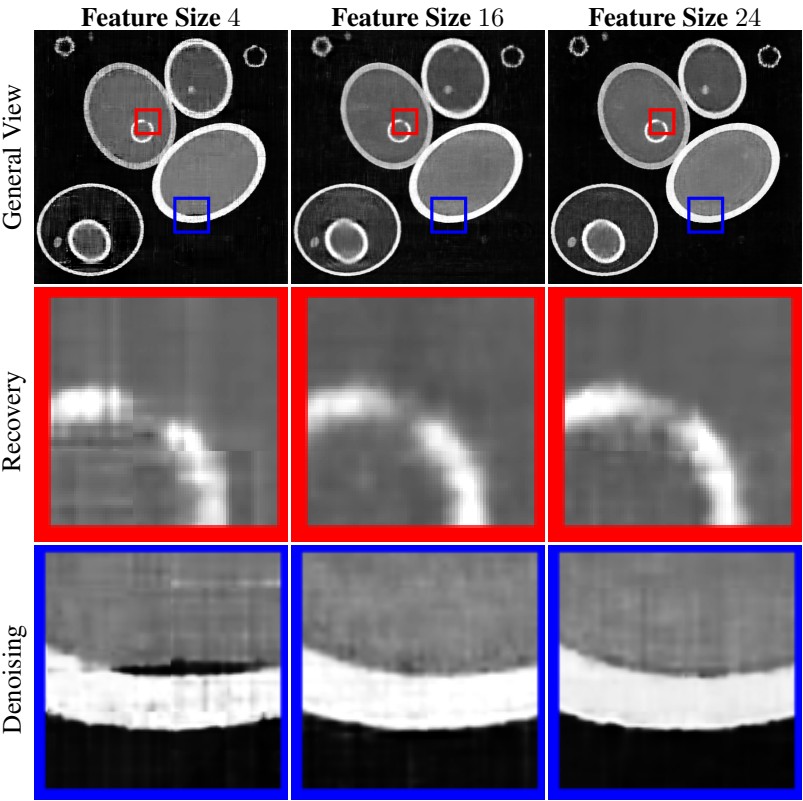

Figure II: Illustration of the impact of the feature size on the reconstruction quality.

## C Additional experiments using real datasets

### C.1 Impact of the Boundary Consistency Constraint (BCC)

To validate the effectiveness of the BCC, we illustrate in the Figure III a slice visualization of the reconstructed tomogram without using the BCC (first row), and the same reconstruction using the BCC prior (second row). One can notice clearly the discontinuity artifacts between adjacent quatree nodes when the BCC is not used. However, the use of BCC ensure consistent borders between adjacent quadtree nodes.

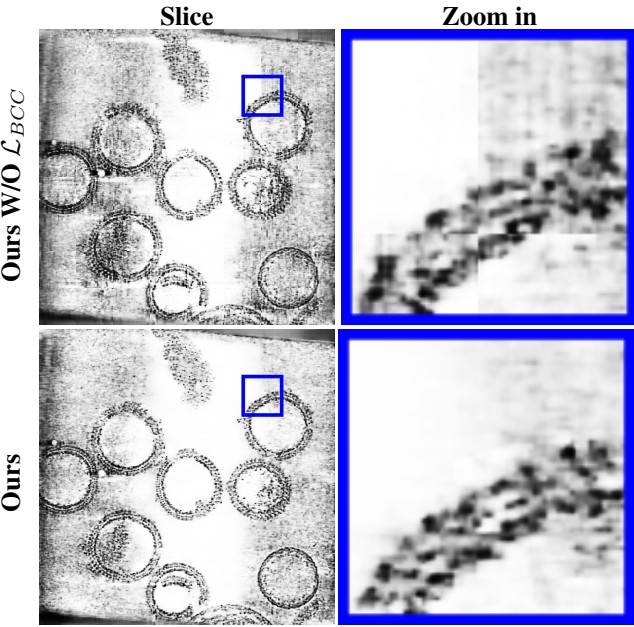

Figure III: Illustration of the impact of the BCC on the reconstruction result.

### C.2 Impact of the Isotropic Fourier Prior (IFP)

In Figure IV, we visualize the impact of the IFP. To make a better comparison, we also add the results obtained with **TensoRF** approach. From top to down, we present the visualization of the reconstructed tomograms for **TensoRF**, **Ours W/O** $\mathcal{L}_{IFP}$ and **Ours**. For the **TensoRF** approach, we can clearly see the structured artifacts. These artifacts are more accentuated on the boundary areas. **Ours W/O** $\mathcal{L}_{IFP}$ has less artifacts, due to the partition methods we used, the boundary artifacts do not impact much on the central region. When we introduce the IFP to the loss function, these artifacts are completely removed. These artifacts are characterized with strong peaks in Fourier domain along the vertical and horizontal directions.

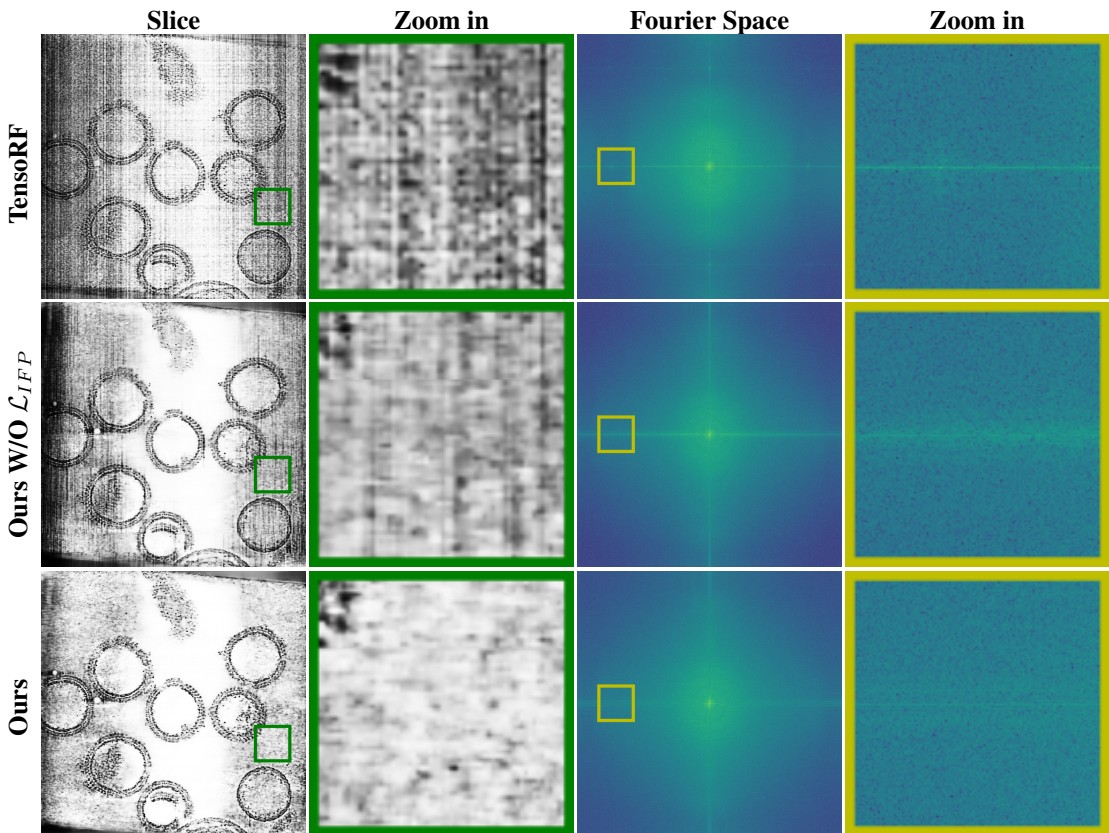

Figure IV: Validataion of the IFP on EMPIAR 10643-40. We also compared with **TensoRF**. **Ours W/O** $\mathcal{L}_{IFP}$ has less artifacts, and **Ours** completely removes the artifacts. The strong peaks in Fourier domain along the vertical and horizontal directions are reduced a lot.

### C.3 Reconstruction results for EMPIAR 10643-51 dataset

As mentioned in the main paper, we also evaluated our reconstruction framework on another serie of the EMPIAR 10643 dataset. We illustrate in the following some of the results that we obtained using this dataset: EMPIAR 10643-51. Figure V shows a comparison of our resonstruction with different baseline methods: (1) **SART+TV**, a well established iterative reconstruction technique **SART** combined with a total variation prior. (2) **Kniesel et al.**, an implicit neural representation for cryo-ET proposed by [2]. (3) **I-NGP**, a reimplementation of Instant-NGP [4] for cryo-ET reconstruction. (4) **TensoRF**, a reimplementation of Tensor Radiance Fields [1] for cryo-ET reconstruction. This comparison shows that our reconstruction yields a less noisy reconstruction while it allows a better details recovery.

We also illustrate in Figure VII, the artifacts removal using the IFP on this dataset.

Furthermore, as for the two datasets presented in the main paper, we conducted a profile analysis shown in Figure VI.

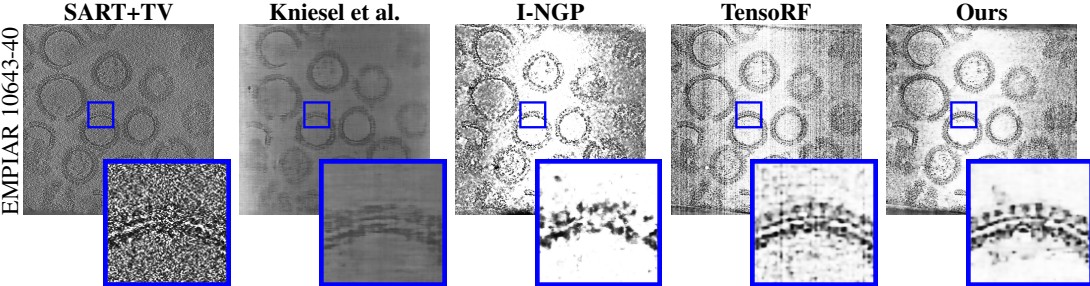

Figure V: Reconstruction results of the HIV-1 (EMPIAR 10643-51).

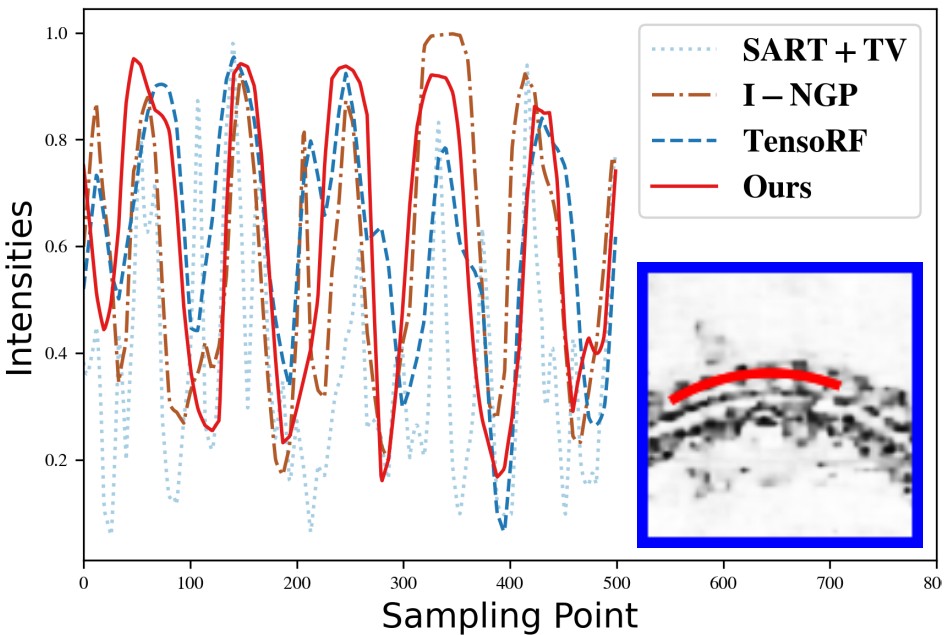

Figure VI: Intensity profile along the red line in the zoomed regions of the reconstruction of EMPIAR 10643-51 dataset.

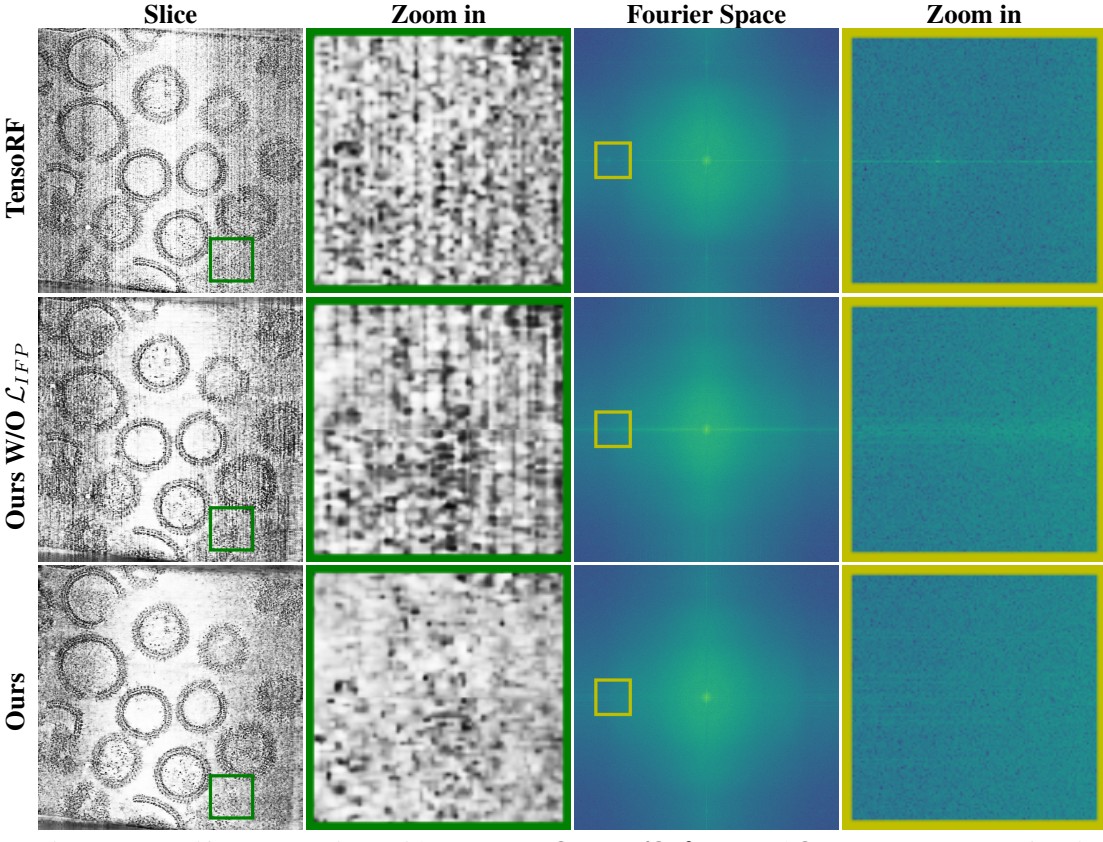

Figure VII: Artifacts comparison with **TensoRF**, **Ours W/O** $\mathcal{L}_{IFP}$, and **Ours** on EMPIAR 10643-51 dataset. **Ours W/O** $\mathcal{L}_{IFP}$ has less artifacts than **TensoRF** approach, while **Ours** completely removes those artifacts.

## C.4 Selection of CNR and ENL regions

In the main paper we introduce the Contrast-to-Noise Ratio (CNR) and the Equivalent Number of Look(ENL) metrics [3] to evaluate the effectiveness of our method. These two metrics are defined as follows:

$$\text{CNR} = \frac{1}{N_{pr}} \sum \frac{\mu_f - \mu_u}{\sqrt{0.5(\sigma_f^2 + \sigma_u^2)}} \tag{2}$$

where $N_{pr}$ is the number of paired regions selected to compute the CNR. $\mu_f$ and $\sigma_f^2$ refer to the mean and variance in the selected regions containing features. $\mu_u$ and $\sigma_u^2$ are the mean and variance in the selected uniform regions. We illustrate these selected regions in the Figure VIII using the red boxes.

$$\text{ENL} = \frac{1}{N_r} \sum \frac{\mu_r^2}{\sigma_r^2} \tag{3}$$

where $N_r$ is the number of regions selected to compute the ENL. $\mu_r$ and $\sigma_r^2$ refer to the mean and variance in the selected homogeneous regions. The selected regions to compute this metric correspond to the purple boxes in Figure VIII.

The CNR evaluates how the denoiser strategy increases the contrast between the region of interest and the background. The ENL metric measures the smoothness in the homogeneous regions.

We manually selected the CNR region pairs and ENL regions, and applied the selection to all the other methods. Here, we chose a 3D volume with a depth of 10 to the rectangle regions to evaluate the reconstructed volume.

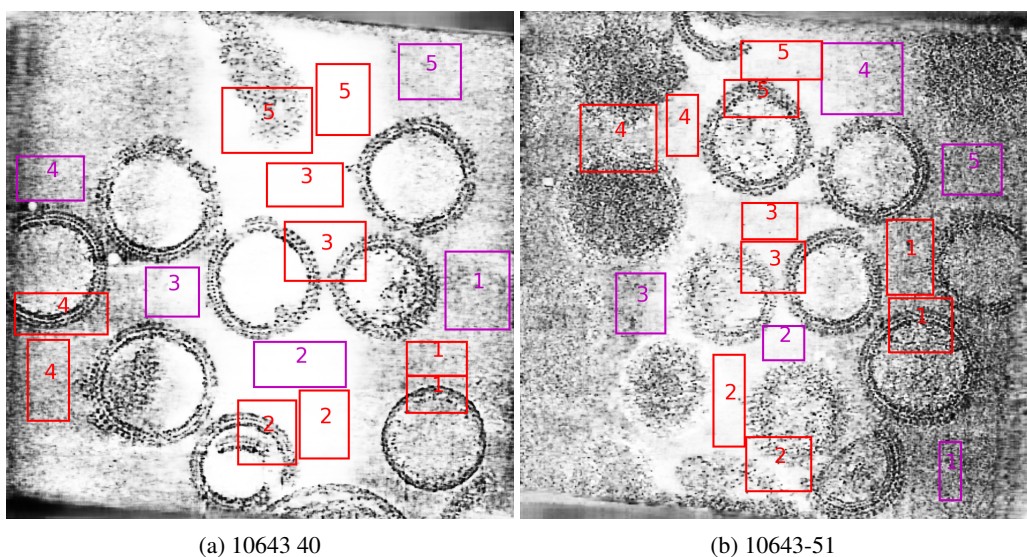

(a) 10643 40        (b) 10643-51

Figure VIII: ENR CNL region choice

## C.5 Slice view for EMPIAR 10643-40 dataset

In the Figure IX, we show a visualization of different slices from our reconstruction of the EMPIAR 10643-40 dataset. One can notice the consistency of the reconstruction over different slices.

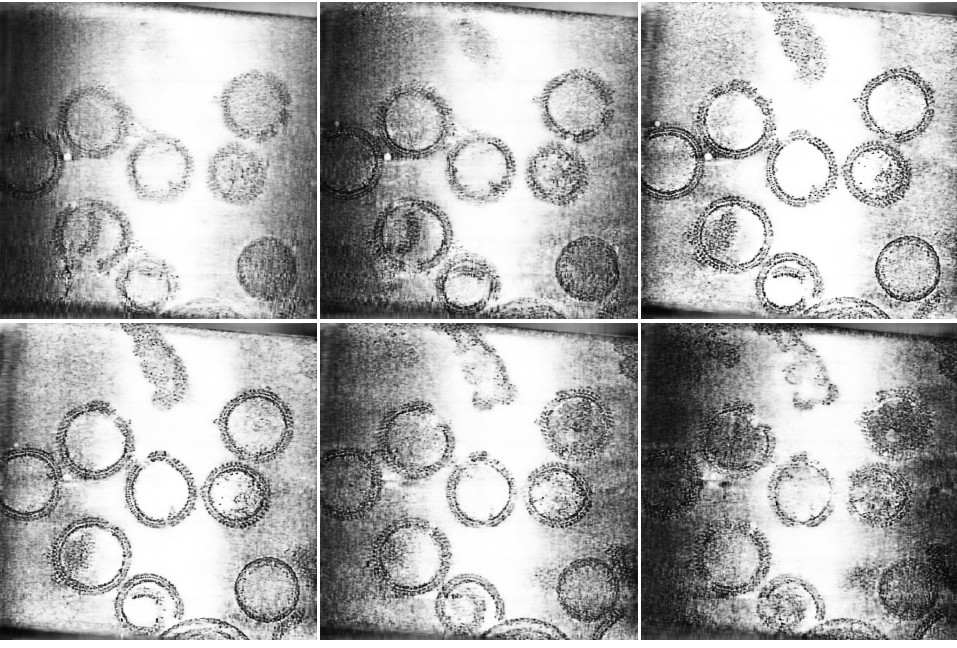

Figure IX: Visualization of different slices from the tomogram obtained when reconstrctinf the EMPIAR 10643-40 dataset.

## D Running time and memory consumption

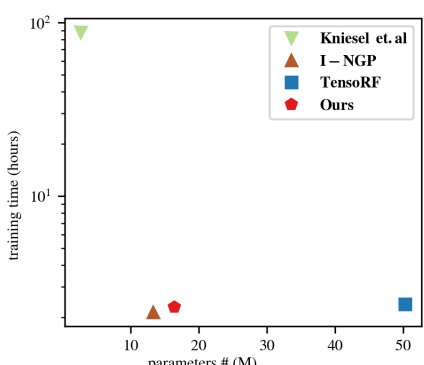

Figure X: Comparison of the number of parameters used and the training time, for the learning-based approaches used in our comparison.

We compare in the Figure X the total execution time needed to perform the training and reconstruction, as well as the number of parameters to be trained in each approach. In this figure, we did not report the running time and parameter number needed in **SART+TV**, as this approach is not a learning-based approach. The running time for **SART+TV** after acceleration is around 4 hours, on our workstation Section A. We can see that **Kniesel et al.** needs fewer parameters, but consumes 40 times more computation time than other approaches. Our approach uses a similar number of parameters than **I-NGP**, and around half in comparison to **TensoRF**, but yields better results as shown in the main paper and the supplementary.

## E 4K-size reconstruction results

We ran our code on the original EMPIAR 100643-40 datasets with $4096 \times 4096$ resolution projections, and visualize the result in this section. For this experiment, we select a tensor dimensions equals to: $180 \times 180 \times 230$. The total number of parameters is evaluated to be around $144.95$ million parameters,

and it took around one day to finalize the reconstruction. Two slices of the resulted tomogram are represented in Figure XI.

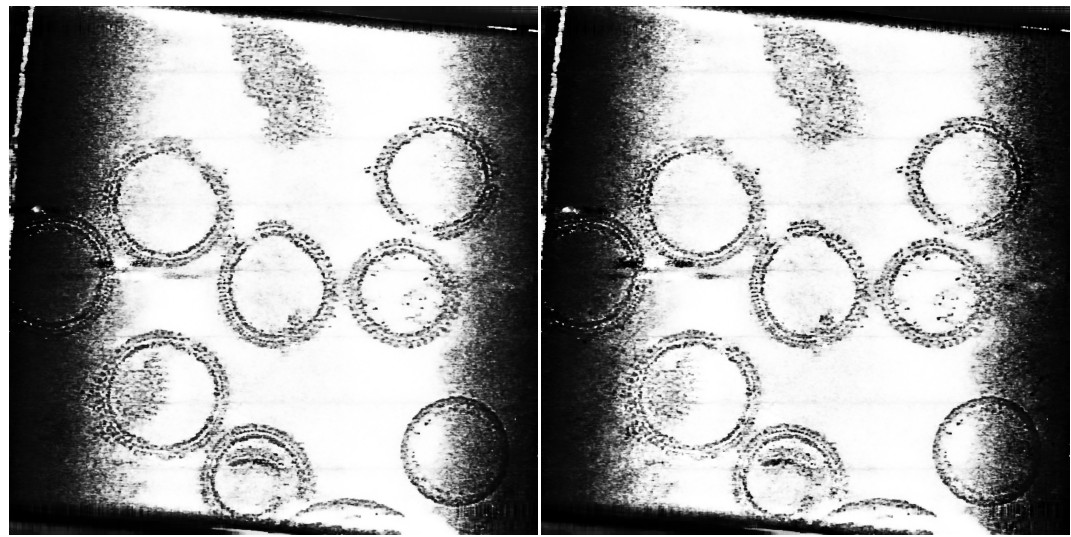

Figure XI: Visualization of two slices from the tomogram obtained using the original 4K projections of the EMPIAR 10643-40 dataset.