# OpenReview forum: "Learning Adaptive Tensorial Density Fields for Clean Cryo-ET Reconstruction"
_NeurIPS.cc/2023/Conference — NeurIPS 2023 poster_

### Official Review · Reviewer_6xkQ · 2023-07-01

**Soundness:** 4 excellent
**Presentation:** 3 good
**Contribution:** 4 excellent
**Rating:** 8
**Confidence:** 4

**Summary:**

The authors address the problems of denoising and tomographic reconstruction, specifically in the context of cryoelectron tomography (cryoET), in which 3D structures (e.g. of proteins) are reconstructed using a tilt-series of tomograms. CryoET holds much promise for the elucidation of biological structures with atomic or near-atomic resolution in situ but faces significant challenges in denoising and reconstruction. While a good deal of work exists to address both of these challenges, the authors here introduce a novel method using tensor density fields that jointly solves the reconstruction and denoising problems, a first. Another innovation is the introduction of a isotropic Fourier regularization term in their loss function, which serves to ameliorate issues with streaking artifacts. In sum, the authors make a significant contribution to cryoET reconstruction with applications to structural biology and other fields in which cryoET is used.

**Strengths:**

Originality: The authors here present the first (to my knowledge) method in which tomographic reconstruction and denoising are jointly solved for cryoET. The authors employ an innovative architecture and pipeline that enables unprecedented computational speedup for large, data-intensive cryoET datasets. Finally, the authors incorporate a novel regularization term (isotropic Fourier prior) that helps deal with artifacts and yields better results than state-of-the-art approaches for cryoET.

Quality: The authors deliver significant improvements on existing methods for denoising, reconstruction, and computation time for cryoET datasets. Their method is validated on both synthetic and real data, demonstrating applicability. The code is efficiently implemented and results in significant improvements to computational efficiency.

Clarity: The paper is written exceptionally clearly. The authors provide extensive information on existing methods for denoising and clearly states the ways in which their method differs and innovates on previous ones. The authors also delineate conditions under which their methods performs better or worse.

Significance: New methods for the effective reconstruction and denoising of cryoET data promises to make this powerful method more accessible, with the potential for making new discoveries in structural biology and other fields. As a method that jointly addresses reconstruction and denoising efficiently, the author's method makes a significant contribution. The method or representational architecture might also be applied to other tomography modalities.

**Weaknesses:**

Clarity: Although minor, some of the figures could be presented more clearly.

Quality: The authors mention that the issues they solve in the context of cryoET are general to many tomographic reconstruction problems. While true in principle, the authors do not demonstrate that their method work well in other modalities compared to state-of-the-art.

**Questions:**

Figures 4 and 7: Increase the font sizes of the axes for better readability. Adjust some of the lighter line colors for better readability.



**Limitations:**

Yes, limitations are clearly addressed.

---

> ### Author Rebuttal · Authors · 2023-08-09
>
> Thank you for you recommending. We thank Reviewer 4 for their valuable feedback, and we will make the necessary changes to the font size and color of the graphs.

---

### Official Review · Reviewer_9Yvf · 2023-07-03

**Soundness:** 3 good
**Presentation:** 3 good
**Contribution:** 3 good
**Rating:** 6
**Confidence:** 3

**Summary:**

This paper proposed a field-based method for 3D image reconstruction in Cryo-EM, which is a challenging task due to strong measurement noise and ill-posedness. The proposed method, **TensorDF**, has the following features: 1) it combines implicit representation and a quadtree, where each node corresponds to a feature tensor, 2) it can automatically update the quadtree structure during training, and 3) it includes three regularizers, that is, total variation, boundary consistency, and penalty in the Fourier space, to improve the reconstruction accuracy.

Experimental validations on both simulated and real datasets are presented. **TensorDF** is compared with several baseline methods and yields superior performance. The authors also discuss how the hyperparameters of TensorDF are selected, and demonstrate the robustness of **TensorDF** under different noise levels.

**Strengths:**

1. Clear presentation of the proposed method.
2. Solid experiments with good performance.
3. Thorough supplementary material (ablation study, more visual examples, and discussion).

**Weaknesses:**

I do not see any obvious weakness.

**Questions:**

1. Could the author further explain the factorization of each feature map $p_i^m$? namely $V_x$, $V_y$, $V_z$, and $M_{xy}$, $M_{yz}$, $M_{xz}$. I am not clear on what is trainable and what is not.

2. Although MLP is not scalable to large volumes, the network itself can help regularize the final reconstruction (see [1]). Perhaps not using an MLP makes the additional regularizers necessary?

3. A followed-up question is how good/bad will TensorDF w/o all regularizers perform? This can show if a quadtree structure can impose some regularization.

[1] Recovery of continuous 3D refractive index maps from discrete intensity-only measurements using neural fields.

**Limitations:**

The limitation of the proposed method is not discussed.

---

> ### Author Rebuttal · Authors · 2023-08-09
>
> a1: All the Vs and Ms are trainable variables. Vs and Ms work like volume factorization. Our target is to optimize the volume, and it is represented as a product of Vs, and Ms. Decoder is also optimized by default.
>
> a2: The reviewer is correct that the absence of the MLP necessitates the additional regularizer. If we use direct explicit expression, we will encounter many local minima, which is why we need to regularize the model. However, explicit representation has the advantage of being fast. Moreover, even with random initialization, it can produce stable reconstructions.
>
> a3: We appreciate the reviewer’s suggestion to perform comparisons to illustrate the quadtree regularization effect in our approach. Indeed, the quadtree structure has more benefits than just speeding up the computation. It also allows us to have more local matrix representations in the XY and XZ planes, compared to TensorDF which can only have global planes for the whole scene. This can reduce the noise and misrepresentation in the reconstruction. That is why our method achieves a significant improvement in quality with only a 50 percent reduction in parameter size. We provide in the rebuttal.pdf file some additional experiments to illustrate the impact of the quadtree as a regularizer. Since TensorDF (TensorRF in the original paper) already incorporates TV prior and does not require boundary consistency prior, we compare our approach to: TensorDF W/O TV prior, TensorDF, and TensorDF W the Isotropic Fourier Prior.

---

### Official Review · Reviewer_tcrC · 2023-07-06

**Soundness:** 2 fair
**Presentation:** 1 poor
**Contribution:** 2 fair
**Rating:** 3
**Confidence:** 5

**Summary:**

In their paper, the authors present a learning-based framework that tackles the challenges faced in reconstructing 3D structures from tilt-series cryo-Electron Microscopy (cryo-EM) data. Cryo-EM is a powerful imaging technique known for its ability to achieve near-atomic resolutions. However, it is not without its drawbacks, including missing-wedge acquisition, large data size, and high noise levels.

To address these challenges, the authors introduce an innovative approach that utilizes an adaptive tensorial-based representation for the 3D density field of the scanned sample. The framework consists of several key components. First, a quadtree structure is optimized to partition the volume of interest effectively. Then, a vector-matrix factorization technique is employed to learn the tensor representation of the density field within each node.

To further enhance the reconstruction quality, the authors incorporate a loss function that combines a differentiable tomographic formation model with three regularization terms: total variation, boundary consistency constraint, and an isotropic Fourier prior. This allows the authors to generate high-quality 3D tomograms and query density information at any location using the learned representation.

The authors demonstrate the superiority of their framework over existing methods using both one synthetic and one experimental dataset. They claim that their framework not only enhances the quality of reconstructions but also reduces computation time and memory footprint. Overall, the paper presents a novel framework that addresses critical challenges and provides potential improvements in cryo-EM reconstruction.


**Strengths:**

*Adaptive Tensorial Representation:* The framework utilizes an adaptive tensorial-based representation for the 3D density field, which should in theory wllow for efficient partitioning of the volume of interest. This adaptive approach ensures that the representation is tailored to the specific characteristics of the sample, enhancing the accuracy of the reconstruction.

*Comprehensive Regularization:* The authors incorporate multiple regularization terms, including total variation, boundary consistency constraint, and an isotropic Fourier prior. This comprehensive regularization scheme should help mitigate the challenges associated with missing-wedge acquisition, large data size, and high noise levels. It should promote smoothness in the reconstructions and improve the overall quality.

*Querying Flexibility:* The learned representation enables querying of the density at any location within the reconstructed volume. This flexibility is valuable for further analysis and examination of specific regions of interest, providing researchers with a detailed understanding of the 3D structure.

**Weaknesses:**

The motivations behind the Coordinate-based representation (CBR) in this study remain unclear. While the compressed nature of CBR suggests it may possess desirable regularizing properties for this application, the specific effects of this representation have not been thoroughly explored by the authors. It remains uncertain whether CBR offers any data efficiency advantages or exhibits any invariances that could benefit the reconstruction process. To gain more insights into the robustness of the CBR representation, it would be beneficial for the authors to conduct ablation studies that combine alternative representations with the proposed loss functions. Such experiments would shed light on the effectiveness and reliability of CBR, further elucidating its potential contributions to the overall framework.

Reproducibility is of concern, since the authors have not provided access to their code or shared example reconstructions. In addition to expanding on the theoretical details of the method (see above), It would be beneficial for them to present more examples beyond just one simulated and one experimental dataset. Including a wider range of examples would enhance the comprehensiveness and generalizability of their findings, providing a more robust evaluation of their method's capabilities.

As an application paper, it is worth considering the suitability of the chosen simulated data in this study. The authors should aim to evaluate their method on a more relevant dataset that provides ground truth information. It is crucial to assess the performance of the proposed method in a scenario where accurate and relevant reference data is available for comparison.

The paper acknowledges challenges such as the missing-wedge problem, but it lacks results that directly address this issue. Furthermore, there is no mention of colored noise or the point-spread function, which are important factors in cryo-EM reconstruction.

The authors assert that their method offers "faster optimization and better feature recovery." However, there is a lack of benchmarks or evidence showcasing the claimed faster optimization. The statement that reconstructions take "less than a day" sounds excessively slow when compared to several existing methods that achieve reconstruction within a few minutes.

To strengthen their claims, the authors should provide specific benchmark comparisons demonstrating the improved optimization speed of their method. Additionally, it would be beneficial for them to address the computational efficiency aspect and compare their results with other methods that achieve faster reconstruction times. This would provide a clearer understanding of the advantages and limitations of their proposed approach.

**Questions:**

1. What are the motivations behind the specific formulation of the Coordinate-based representation (CBR) in this study? What are the data efficiency advantages or invariances exhibited by CBR that could benefit the reconstruction process?
2. Could the authors present more examples beyond one simulated and one experimental dataset to enhance the comprehensiveness and generalizability of their findings?
3. How does the method address challenges like the missing-wedge problem, colored noise, and the point-spread function?
4. Can the authors provide benchmarks or evidence demonstrating the claimed faster optimization?
5. What is the reason behind representing the "one-layer decoder network" (\mathcal{D}) as a function instead of a matrix multiplication? Could you provide clarity on its parameter size?
6. Could you please explain the definitions of the Vs and Ms variables in equation (4)? Additionally, what is the output dimensionality of these variables?
7. Can you clearify which parameters are trained?
8. Can you provide information on the memory footprint of the proposed framework? How does it impact computational resources and efficiency?

**Limitations:**

1. Lack of clarity and exploration of the motivations and effects of the Coordinate-based representation (CBR).
2. Absence of access to code and example reconstructions, affecting reproducibility.
3. Insufficient number of examples, limiting the comprehensiveness and generalizability of the findings.
4. Potential limitation in the suitability of the chosen simulated data, emphasizing the need for evaluation on more relevant datasets with ground truth information.
5. Failure to directly address challenges like the missing-wedge problem, colored noise, and the point-spread function in cryo-EM data.
6. Lack of benchmarks or evidence supporting the claim of faster optimization in comparison to existing methods.
6. Need for specific benchmark comparisons and a focus on computational efficiency to substantiate claims and understand the advantages and limitations of the proposed approach.

---

> ### Author Rebuttal · Authors · 2023-08-09
>
> a1: Recently, several works [1,2,3,4] have demonstrated the superiority of Coordinate-based representations in solving tomographic problems, especially in missing-wedges scenarios, compared to traditional methods. So there is excellent evidence for the superiority of CBNs in that regard. However, existing approaches are aimed at X-ray projection data, and do not deal with the high level of noise that we encounter in Cryo-ET reconstruction, which is the main issue we want to address in this paper. In our comparison, we selected SART+TV as the baseline of traditional approaches since it performs better than the widely used Weighted Filtered Backpropagation (WFBP) [5]. Our comparison shows better reconstruction results for our approach than SART+TV. Moreover, this traditional approach requires at least 128G (no matter CPU or GPU) for reconstructing a 4K size dataset, while we can run our CBR-based method on a 48G GPU with less CPU memory usage.
>
> a2: We conducted experiments on one simulated data and two real datasets in the main paper: EMPIAR 10643-40 (HIV-1 viruses) and EMPIAR 10751 (HEK cell). In the supplement, we added more experiments on another series of HIV-1 viruses (EMPIAR 10643-51). These datasets are representative of the typical data encountered in structural biology, with respect to noise distribution and feature structures. With our approach, learning is performed for each single dataset, so we do not face overfitting or generalization issues in this case. However, we will include more experiments with new datasets in the supplement to further demonstrate our method. Moreover, it is important to note that cryo-ET does not provide data with ground truth information, which limits our validation options.
>
> a3: [3,4] prove that CBR-based methods could deal with the missing-wedge problem quite well compared with SART+TV and FBP. The point-spread function (PSF), or contrast transfer function (the Fourier transform of the PSF), is corrected during the preprocessing step using IMOD. For some datasets in EMPIAR databank, the CTF is already corrected and the projections are aligned. In the revised version we will make it clear that the CTF should be corrected as a preprocessing step. We also performed some experiments using the approach of [6] to compensate somehow the CTF, but we did not get any clear improvement with our datasets. This means that the preprocessing resulted in a good CTF correction. By running our approach on real captured data from EMPIAR databank, we have shown that the reconstructed tomogram is less noisy than the results from the baseline approaches. Specifically we computed two metrics (CNR and ENL), that evaluate the contrast improvement and the denoising effect. We also evaluate the intensity profiles along a line that contains virus spikes, to show that our approach has a better distinction between spikes and background. In summary, these experiments show that our approach does a better job dealing with the colored noise in the real data.
>
> a4: We test the optimization time using different methods. Ours and I-NGP converge in a similar time (around 2 hours for the 1K dataset). [6] takes several days. SART+TV needs around 4 hours, with a Tomosipo [7] based implementation. Please refer to Part C of the supplementary.
>
> a5: The decoder network consists of a single layer that applies a non-linearity function to the model. This non-linearity enhances the performance of the model in comparison to simple matrix multiplication, as demonstrated by [8] for similar tasks. The decoder has only 0.005 M parameters, which is much smaller than the total number of 28.2 M parameters in the model.
>
> a6: Vs and Ms correspond respectively to rank-one (vector) and rank-two (matrix) tensor components. The output dimensionality of these variables depends on the dimension decomposition. For example, for a scene with the following dimensions: n * m * p, Ms^(X,Y) has a dimensionality of n * m, while Vs^(Y) has a dimensionality of m. We will add the dimensionality information in the revised version.
>
> a7: In our framework, we optimize the vector and matrix factors Vs and Ms for all nodes, as well as the decoder D.
>
> a8: We only consider the GPU memory here. It depends on the batch size setting (the number of samples processed before the model is updated). A larger batch size usually requires more memory, but also speeds up the computation. Therefore, we try to use a relatively large batch size to fully utilize the GPU memory capacity, such as 44GB out of 48GB available on our cards.
>
> [1]Sun, Yu, et al. "Coil: Coordinate-based internal learning for tomographic imaging." IEEE Transactions on Computational Imaging 7 (2021): 1400-1412.
>
> [2]Liu, Renhao, et al. "Recovery of continuous 3D refractive index maps from discrete intensity-only measurements using neural fields. Nat Mach Intell 4, 781–791 (2022).
>
> [3]Rückert, Darius, et al. "Neat: Neural adaptive tomography." ACM Transactions on Graphics (TOG) 41.4 (2022): 1-13.
>
> [4]Zang, Guangming, et al. "IntraTomo: self-supervised learning-based tomography via sinogram synthesis and prediction." ICCV 2021.
>
> [5]Li, Lun, et al. "Compressed sensing improved iterative reconstruction-reprojection algorithm for electron tomography." BMC Bioinformatics 21.6 (2020): 1-19.
>
> [6]Kniesel, Hannah, et al. "Clean implicit 3d structure from noisy 2d stem images." CVPR 2022.
>
> [7]Hendriksen, Allard A., et al. "Tomosipo: fast, flexible, and convenient 3D tomography for complex scanning geometries in Python." Optics Express 29.24 (2021): 40494-40513.
>
> [8]Karnewar, Animesh, et al. "Relu fields: The little non-linearity that could." ACM SIGGRAPH 2022 Conference Proceedings. 2022.

---

> > ### Comment · Reviewer_tcrC · 2023-08-19
> >
> > Thank you for your rebuttal.
> >
> > a1. Kindly address question 1 with a response that directly pertains to the inquiry. While presenting improved results, it remains crucial to provide theoretical underpinnings for your assertions due to the substantial limitations evident in your evaluation.
> >
> > a2. "so we do not face overfitting or generalization issues in this case" It's absolutly possible to overfit to the data, leading to a representation and mapping that might not generalize to held-out data in each datasets. The authors need to consider how this affects the quality of the learnt representation. "cryo-ET does not provide data with ground truth information" Considering this, the incorporation of a more representative synthetic dataset becomes imperative. Please refer to my initial review on this topic.
> >
> > a3. The ability to fully correct for CTF zero-crossings is restricted, emphasizing the necessity of elucidating how your proposed approach manages this challenge. Given the current shortfall in experimental evaluation, it's essential to establish a theoretical foundation showcasing how your method addresses the concerns outlined in question 3.
> >
> > In light of the authors not having sufficiently addressed the core points raised in the initial review, my initial score remains unchanged. The primary issues revolve around the lack of theoretical substantiation for the presented claims and the absence of comprehensive experimental assessment.

---

> > > ### Author Response · Authors · 2023-08-19
> > > **Answer to reviewer tcrC additional concerns**
> > >
> > > Thank you for your time and reply. We assume that our rebuttal has addressed most of the concerns (5+ of 8) in these fields, evidence in faster optimization, etc.
> > >
> > > a1. We hope we addressed the data efficiency advantages in question 1. Compared with traditional representations, CBR methods provide a continuous mapping from the coordinates to the densities. As in many other fields of machine learning, theoretical models for neural fields lag far behind the state of the art of methods deployed in practice. We do not believe that this constitutes reasonable grounds for rejecting work, especially in the face of overwhelming evidence for the performance of neural fields, demonstrated by both our experiments and the mentioned earlier works on neural fields for missing wedge tomography, mentioned in the first rebuttal and cited in the paper. Besides, we evaluate our methods in different metrics, like 3D PSNR and 3D SSIM in the synthetic dataset, CNR, ENL, and profile analysis in real datasets in the paper. We also analyze the main parameters, like the tensor dimensions and feature size.
> > >
> > > a2. We use the synthetic dataset from [1], which also targets the cryo-ET reconstruction using neural representation, and consider random densities and shapes for cryo-ET simulation. It could help distinguish between different methods. In this regard, this data is representative of the features and noise levels found in real cryo-ET data. In addition, we took three real datasets to verify our methods.
> > >
> > > a3. As mentioned in the original rebuttal, we treat the CTF correction as a preprocessing step to our method, using established tool chains routinely used in cryoET reconstruction. Specifically, we use IMOD [2] for this step. Please refer to [3] for analytical and theoretical analysis.
> > >
> > > [1] Kniesel, Hannah, et al. "Clean implicit 3d structure from noisy 2d stem images." CVPR 2022.
> > >
> > > [2] Mastronarde, David N., and Susannah R. Held. "Automated tilt series alignment and tomographic reconstruction in IMOD." Journal of structural biology 197.2 (2017): 102-113.
> > >
> > > [3] Xiong, Quanren, et al. "CTF determination and correction for low dose tomographic tilt series." Journal of structural biology 168.3 (2009)

---

### Official Review · Reviewer_bANQ · 2023-07-06

**Soundness:** 3 good
**Presentation:** 3 good
**Contribution:** 2 fair
**Rating:** 5
**Confidence:** 4

**Summary:**

The work combines quad-tree structure with low-rank tensorial representation to adaptively model cryo-EM volumetric density for reconstruction problem. The quad-tree structure is updated by merging or splitting to encourage uniformity in the area of each node, while the feature tensors for each node are obtained by outer product of learnable vectors and matrices. For the optimization, three regularization terms are used: total variation on vector and matrix features, boundary constraints to ensure points on the edge of two neighboring nodes have similar features, and Isotropic Fourier Prior. The latter penalizes outliers in horizontal/vertical Fourier coefficients and thus mitigates axis-aligned artifacts. A two-stage, coarse-to-fine optimization with downsampled projects helps avoid overfitting to noise.
Through experiments on synthetic data, effect of tensor and feature dimensions as well as robustness to noise is studied. On real data, the proposed method is compared with SOTA and related implicit representation learning.

**Strengths:**

1. Introduction to the problem is well written. It perfectly provides necessary background to understand the challenges in tilt-series tomography.
2. The work combines spatial partitioning idea with tensorial based representation to obtain a compact representation of the density field.
3. Two simple yet effective regularizers of Boundary constraint and Isotropic Fourier are cleverly used to mitigate artifacts and their impact is supported with evidence.

**Weaknesses:**

1. Based on my understanding, the proposed method is building upon TensoRF by locally (rather than globally) defining tensorial representation in an adaptive quad-tree structure, while adding regularizations to avoid artifacts and impose smoothness. Some ablation studies are missing examining the sole effect of each of these additions, e.g. qualitative and quantitative results on TensoRF (global representation) with all the new regularization terms. Currently, it is not clear how much of the improvement in denoising metrics (or qualitative results) is because of quad-tree vs. regularization terms. In other words, quad-tree is claimed to be important contribution but its effect is not fully studied.
2. The isotropic Fourier regularization term requires Fourier Transform which can be expensive and slows down the computation, although a coarse sampling helps reducing the computation.

**Questions:**

1. According to L229-233, the optimization is divided into two stages. In the first stage, both the quad-tree and tensorial representations inside each node are updated. The question is: when a node subdivided, how are tensorial representations of the new nodes initialized? Are they related to the tensorial representation of the parent node? What about the case when some nodes are merged together?
2. How much of the optimization time in the first stage is spent on quad-tree updates vs. tensorial representation updates? How expensive is this discrete optimization problem? How do you set the hyper-parameters, such as the threshold for STD, for this step?

**Limitations:**

Apart from above questions and concerns, I have a suggestion: you might be able to replace the Isotropic Fourier Prior with another one that can be computed more efficiently in real space (rather than Fourier space).

---

> ### Author Rebuttal · Authors · 2023-08-09
>
> a1: In our implementation, we start with a fixed number of nodes (for example, 70, as mentioned in the paper), we initialize each tensor representation with random values between -0.5 and 0.5. After each iteration, some nodes may become inactive, because they are subdivided or merged with others. Then, we map the currently active nodes to the initial 70 nodes. This mapping has no geometrical considerations. Thus, the tensor representations from one iteration to another (of the quadtree update) are not related. However, after only a few iterations of the tensor representations optimization we converge effectively, demonstrating our representation's robustness to initialization.
>
> a2: In the first stage of optimization, around 10% of the total optimization time is dedicated to quadtree optimization. Then, we fix the quadtree structure and optimize the tensorial representation. This step accounts for around 85% of the total optimization time. The STD calculation takes around 5 minutes and represents less than 5% of the overall computation time. The discrete optimization takes less than 5 seconds. We do not use a threshold for STD, and the updating takes into account the STD for all current nodes to decide which should be divided, merged or kept the same. However, we impose a limit on the total number of nodes (we set 70 in the paper). We also add discrete constraints similar to [1]:
>
> We define three binary variables for each node: ws, wk, and wm, which indicate whether the node should be subdivided, kept, or merged.
> We ensure that the sum of these variables is one for each node.
> We enforce the total number of nodes to not exceed a fixed limit.
> Then, we minimize the objective function, which is the sum of the products of the binary variables (ws, wk, and wm) and the STD of each node. We solve this problem using mixed integer programming with or-tools [2].
> [1]Julien N. P. Martel, David B. Lindell, Connor Z. Lin, Eric R. Chan, Marco Monteiro, and Gordon Wetzstein. Acorn: Adaptive coordinate networks for neural scene representation. ACM Trans. Graph., 40(4), 2021
> [2]Laurent Perron and Vincent Furnon. Or-tools.
>
>
> answer to limitations: We appreciate your insight and will keep it in mind for our future work. We agree that using a feature extractor or a filter in real space could be beneficial for our design. We tried using a pre-trained VGG model as a feature extractor, but it did not work well because it was trained on ImageNet, which differs from our dataset and feature space. The Isotropic Fourier Prior does not consume much memory after our optimization. We use some techniques to reduce memory usage, such as releasing the memory of the intermediate computations and sampling the strategy. The computation time is still relatively high, but we think it is worth it for improving the quality of the results.

---

> > ### Comment · Reviewer_bANQ · 2023-08-21
> >
> > Thank you for answering the questions. Unfortunately, I see no reply to discussed weaknesses so concerns are remained for validation of choices made in the design of the method. I believe the work at this current stage lacks suggested ablation studies. I should decrease my score to Borderline Reject.

---

> > > ### Author Response · Authors · 2023-08-21
> > > **Replying to Reviewer bANQ**
> > >
> > > Dear bANQ,
> > >
> > > Thank you for your kind reminding. Actually, we did the experiments regarding your first weakness concern in the first round review, 'TensoRF (global representation) with all the new regularization terms'. It is listed in reply to Reviewer 9Yvf, and the results are attached with rebuttal.pdf in the main rebuttal reply. The questions are similar. We are sorry that we did not specifically reply here. It would be appreciated if you could check the results for further consideration.
> > >
> > > Regards,
> > >
> > > Here we attached the reply for question 3 from Reviewer 9Yvf.
> > >
> > > 'a3: We appreciate the reviewer’s suggestion to perform comparisons to illustrate the quadtree regularization effect in our approach. Indeed, the quadtree structure has more benefits than just speeding up the computation. It also allows us to have more local matrix representations in the XY and XZ planes, compared to TensorDF which can only have global planes for the whole scene. This can reduce the noise and misrepresentation in the reconstruction. That is why our method achieves a significant improvement in quality with only a 50 percent reduction in parameter size. We provide in the rebuttal.pdf file some additional experiments to illustrate the impact of the quadtree as a regularizer. Since TensorDF (TensorRF in the original paper) already incorporates TV prior and does not require boundary consistency prior, we compare our approach to: TensorDF W/O TV prior, TensorDF, and TensorDF W the Isotropic Fourier Prior.'

---

> > > > ### Comment · Reviewer_bANQ · 2023-08-21
> > > >
> > > > Thanks for the clarification. I will readjust my score now.

---

> > > > > ### Author Response · Authors · 2023-08-22
> > > > >
> > > > > Thank you, and we hope we could resolve all your concerns.

---

### Author Rebuttal · Authors · 2023-08-09

We include in the rebuttal.pdf file some additional experiments to answer question 3 of Reviewer 3.

---

> ### Author Response · Authors · 2023-08-21
> **This is the result for TensoRF w/o priors comparison**
>
> Dear Reviewers,
> We attached the results for the TensoRF w/o priors and new priors comparison with our methods here.

---

### Decision · Program_Chairs · 2023-09-21

**Decision:**

Accept (poster)

**Comment:**

This paper presents a learning-based method for reconstruction of cryo-ET tomograms. The method combines quad-tree structure with low-rank tensorial representation to adaptively model cryo-EM volumetric density. Several regularization terms are introduced to cope with the high amount of noise in cryo-ET data. The method is innovative and the manuscript shows good empirical results on both synthetic and real data. Acceptance is therefore recommended.